# Dynamics of cooperation in concurrent games

Charlotte S. L. Rossetti [1,2] ✉, Oliver P. Hauser [3,4] & Christian Hilbe [1,4] ✉

People frequently encounter situations where individually optimal decisions conflict with group interests. To navigate such social dilemmas, they often employ simple heuristics based on direct reciprocity: cooperate when others do and cease cooperation when partners defect. However, prior research typically assumes that individuals only interact in one game at a time. In reality, people engage in multiple games concurrently, and the outcome of one interaction can influence behavior in another. Here, we introduce a theoretical framework to study the resulting cross-over and spill-over effects. Participants repeatedly engage in two independent stage games, either with the same or different partners, adapting their strategies over time through an evolutionary learning process. Our findings indicate that individuals often link their behavior across games, particularly under cognitive constraints like imperfect recall. A behavioral experiment with 316 UK-based students suggests that concurrent games negatively affect cooperation, highlighting how strategic motives and spillovers impact reciprocity.

Direct reciprocity is one of the core mechanisms enabling cooperation among unrelated individuals[1,2]. This mechanism is at work when neighbors take turns picking up each others' children from school, when students correct each others' work, or when couples share domestic chores. Experimental work shows that reciprocal relationships emerge naturally if interactions occur repeatedly, provided the probability of another encounter is sufficiently high[3,4]. Repetition allows individuals to condition their current actions on their interaction partner's past behavior[5]. When they adopt conditionally cooperative strategies such as Tit-for-Tat[6–8], Generous Tit-for-Tat[9,10], or generalizations thereof[11–16], even selfish opponents have an incentive to cooperate. Using models of evolutionary game theory, researchers have explored which kinds of strategies evolve, and in which environments reciprocal cooperation is stable[17–24].

Yet most of this work assumes that individuals either only engage in one repeated game at a time, or that they treat each game as independent. This means that both theoretically and experimentally, each ongoing strategic interaction is studied in isolation[3]. This assumption of independence greatly facilitates a theoretical analysis. It allows researchers to consider a comparably small set of possible strategies[25].

Once this assumption is dropped, a player's strategy does not only depend on the opponent's previous actions in the respective game anymore. Instead, it may depend on the previous actions of all opponents, across all games. As a result, the cooperation dynamics need to be described at a different level: instead of the standard game-perspective, models now need to take a population-perspective. This change in perspective drastically increases a model's computational complexity[26]. To circumvent these difficulties, most research is based on the implicit assumption that by analyzing different games individually, one can extrapolate (or at least approximate) how people behave when they engage in many games in parallel. Our aim is to explore to which extent this assumption is justified. We make two key contributions. First, we refute, both theoretically and experimentally, that people generally treat their different games as independent. Second, by taking into account the linkage between games, we introduce a theoretical framework that gives rise to a richer and more realistic class of game-theoretic models.

Our inquiry is based on the notion of a concurrent game. A concurrent game arises when players engage in several, formally independent, repeated games in parallel. Players may have their different

[1]Max Planck Research Group on the Dynamics of Social Behavior, Max Planck Institute for Evolutionary Biology, 24306 Plön, Germany. [2]Department of Psychology, University of Zürich, 8050 Zürich, Switzerland. [3]Department of Economics, University of Exeter, Exeter EX4 4PU, UK. [4]These authors contributed equally: Oliver P. Hauser, Christian Hilbe. ✉e-mail: rossetti@evolbio.mpg.de; hilbe@evolbio.mpg.de

repeated games either with the same or with different interaction partners (Fig. 1). We ask to which extent behavior in the concurrent game can be inferred from the constituent repeated games. This question has received some attention before, as reviewed in detail in the Supporting Information. However, respective models often take a static equilibrium approach[27,28]. This research shows, for example, that if players implement an equilibrium for each isolated repeated game, the resulting strategy profile also constitutes an equilibrium of the concurrent game. When all games are identical, symmetric, and played with the same partner, one can even derive a stronger result. In that case, full cooperation is feasible in the concurrent game if and only if it is feasible in each repeated game[27]. These studies greatly illuminate which behaviors are possible in equilibrium. Yet, they do not address which of these equilibria (if any) are most likely to emerge when strategies are not consciously chosen but learned over time. Moreover, this existing work does not attempt to study the consequences of several cognitive constraints and behavioral heuristics that might affect human play in concurrent games. For example, effects arising from imperfect recall[29–31] or from a drive to act consistently may naturally introduce spillovers between games. Once such spillovers occur, behavior may spread from one game to another[32]. Herein, we study a simple but comprehensive theoretical framework to describe these effects.

We consider three idealized scenarios, to which we refer as treatments. In all treatments, players engage in two different repeated social dilemmas. The two dilemmas either result in a high or a low benefit of cooperation (Fig. 1a). The three treatments differ in whether or not players treat each repeated game as independent, and in whether or not the two games are played with the same or with different interaction partners. In the first treatment, the *control*, we consider the baseline case typically studied in the literature (Fig. 1b). Here, individuals play each repeated game in isolation. Hence they treat each repeated game as independent by design. Second, in the *same-partner treatment*, the two games are played simultaneously, and with the same opponent (Fig. 1c). This treatment is motivated by the previous work of Donahue et al.[25]. They refer to this setup as a 'multichannel game', because players can interact and influence each other through multiple channels. As a result, players can react to an opponent's defection in one game by defecting in the other. In this way, we aim to capture players' strategic motives to link their behavior across different games. This linkage may provide players with a stronger leverage to enforce cooperation. Third, in the *different-partners treatment*, individuals play the two games simultaneously but with a different co-player in each game (Fig. 1d). This treatment is motivated by the study of Reiter et al.[26]. However, in their work, behavioral spillovers occur with an exogenously determined probability. In contrast, we explore whether such linkage would evolve endogenously. By combining a different-partners design similar to Reiter et al.[26], and the same-partner design by Donahue et al.[25], we explore how strategic motives contribute to the evolution of cross-game effects. We further expand this

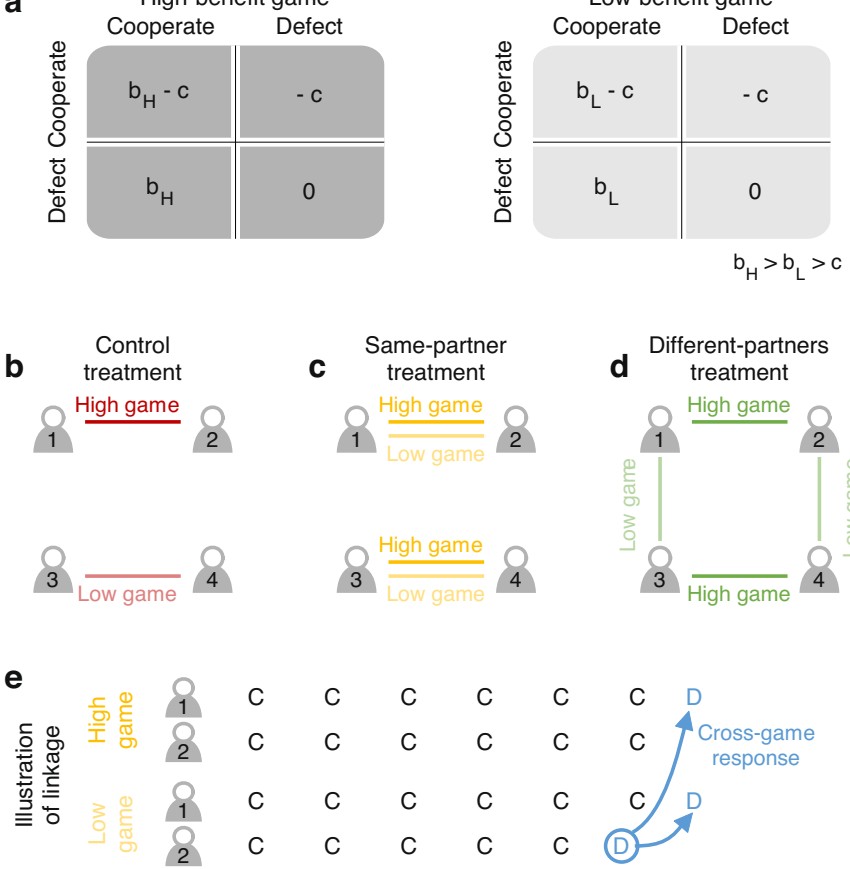

**Fig. 1 | A framework of concurrent games. a** In concurrent games, players engage in two or more games simultaneously. Herein, we consider the case that players engage in two games, one with a high benefit of cooperation ('high game', darker shade) and one with a smaller benefit ('low game', lighter shade). Each payoff matrix describes the payoff of the player who picks a row, depending on the co-player's choice of who picks a column. **b** In the control treatment (red), players only engage in one repeated game at a time, as usually assumed in the literature. **c** In the same-partner treatment (yellow), each player engages in both games but with the same co-player. **d** In the different-partners treatment (green), players engage in both games but with different co-players. **e** Concurrent games allow for linkage. Players might respond to a co-player's defection in one game by defecting in both games. Such linkage may arise both in the same-partner treatment (depicted here) and in the different-partners treatment. C and D represent the decision to cooperate or to defect, respectively.

work by comparing a single game control with the other treatments, exploring in which case players treat each game as independent. In addition, we expand on this previous work by exploring the impact of several plausible cognitive constraints and behavioral heuristics. In this way, we wish to establish a realistic framework to study the evolution of strategic behavior in concurrent games.

In the same-partner treatment, and to a far lesser extent in the different-partners treatment, we find that a player's behavior in one game is linked to the previous outcome of the other game. This linkage can either result in more or less cooperation compared to the control, depending on the treatment and the presence of cognitive constraints. To further explore these theoretical results, we run a behavioral experiment that implements our three treatments, based on a similar design as in previous empirical studies[33,34]. In the experiment, both the same-partner treatment and the different-partners treatment resulted in less cooperation than the control. For the same-partner treatment, our empirical data does not only rule out that players treat each game as independent. It also calls into question a previous prediction by Donahue et al.[25] that concurrently ongoing games among the same partners would enhance cooperation.

Our results have important implications for the effectiveness of direct reciprocity. People in their daily lives often engage in several games concurrently. For such concurrent games, we find that strategic motives, spillovers, and cognitive constraints can easily affect, and often undermine, cooperation.

## Results

### A model of concurrent games

To establish formalism, we first study cooperative interactions based on a variant of the prisoner's dilemma, the donation game[35]. In this game, players either cooperate (C) or defect (D). Cooperation means to pay a cost $c$ for the partner to get a benefit $b$. Defection means to pay no cost and for the partner to get no benefit. We consider two different implementations of this game (Fig. 1a). In one implementation, the benefit is high, and we accordingly speak of the high-benefit game, or *high game* (H). In the other implementation, the benefit is smaller, and we call it the *low game* (L). Assuming $b_H \geq b_L > c$ throughout, the dominant action if players only meet once is to defect in either game. However, we assume players interact for infinitely many rounds (an extension to finitely repeated games will be discussed later). We refer to each iterated donation game as a *repeated game*. When players engage in both donation games in parallel, such that players make two choices each round (one for each game), we speak of a *concurrent game*. A large literature shows that cooperation is feasible in repeated games[5]. This result naturally extends to concurrent games. Here, we are interested in how likely cooperation is to evolve in concurrent games, and which strategies are used to sustain it.

To this end, we discuss three different idealized scenarios (treatments) of how these games unfold. In each case, we consider four players. In the *control treatment*, players only engage in a single repeated game at a time, with a fixed partner (Fig. 1b). One pair of players repeatedly engages in the high game, whereas the other pair plays the low game. Players use reactive strategies to make their decisions. This means that a player's choice of whether or not to cooperate in a given round only depends on the co-player's decision in the previous round. Reactive strategies take the following form[35]:

$$\mathbf{p} = \left( p_C^k, p_D^k \right) \in [0,1]^2. \tag{1}$$

Here, $p_a^k$ is the player's probability to cooperate in game $k \in \{H, L\}$, depending on the co-player's previous action $a \in \{C, D\}$. For example, a player with strategy $\mathbf{p} = (1, 0)$ implements Tit-for-Tat (TFT). A player with $\mathbf{p} = (1, p_D)$ and $0 < p_D < 1$ uses Generous Tit-for-Tat GTFT (see refs. 9,10). Finally, a player with $\mathbf{p} = (0, 0)$ defects unconditionally (ALLD).

We contrast this control treatment with two different kinds of a concurrent game. In the first one, the *same-partner treatment*, players are again matched with a single partner, but the two players interact in both repeated games simultaneously (Fig. 1c). In particular, their decision in either game may depend on how the co-player acted in the other game. Reactive strategies for the same-partner treatment take the form

$$\mathbf{p} = \left( p_{CC}^H, p_{CD}^H, p_{DC}^H, p_{DD}^H; p_{CC}^L, p_{CD}^L, p_{DC}^L, p_{DD}^L \right) \in [0,1]^8. \tag{2}$$

Here, $p_{a^H a^L}^k$ is the player's probability to cooperate in game $k \in \{H, L\}$, depending on the co-player's previous decisions in both the high and the low game, $a^H, a^L \in \{C, D\}$. We say such a strategy treats both games as independent if the entries satisfy

$$p_{CC}^H = p_{CD}^H, p_{DC}^H = p_{DD}^H \quad \text{and} \quad p_{CC}^L = p_{DC}^L, p_{CD}^L = p_{DD}^L. \tag{3}$$

That is, the strategy only reacts to the co-player's previous action in the respective game, irrespective of the outcome of the other game. In the case of independence, strategies of the control treatment naturally map to strategies in the same-partner treatment. For example, if a player in the control were to use TFT in the high game and ALLD in the low game, that player could implement $\mathbf{p} = (1, 1, 0, 0; 0, 0, 0, 0)$ in the concurrent game. Thus, the same-partner treatment permits all strategic behaviors that are feasible in the control. In general, however, the set of feasible strategies is strictly larger in the same-partner treatment. For example, players with $\mathbf{p} = (1, 0, 0, 0; 1, 0, 0, 0)$ only cooperate in either game if the co-player previously cooperated in *both* games. When the constituent games are not treated as independent, we say players link their behavior across games. Accordingly, we also speak of linkage. Examples like the one above illustrate that linkage might arise because of strategic motives. By doing so, players may be able to enforce cooperation more effectively, by threatening to defect in both games after any deviation of the co-player (Fig. 1e).

The last treatment is the *different-partners treatment*. Here, players again engage in both the high and the low game simultaneously, but now with different co-players (Fig. 1d). Reactive strategies for this treatment have the same complexity as in the same-partner treatment, see Eq. (2). Also the definition of independence is the same, see Eq. (3). From a strategic viewpoint, however, this treatment differs from the same-partner treatment. With different partners involved, there is less of an immediate strategic motive to link behavior across games, unless players wish to adopt a strategy of community-enforcement[36,37].

For all three treatments, we can compute the players' payoffs explicitly. To this end, we represent the interaction as a Markov chain that depends on the players' strategies. We describe the respective procedure in the Methods and in the Supporting Information. However, we do not regard the players' strategies as fixed. Rather, as usual in evolutionary game theory, players update their strategies over time based on their payoffs. To model this updating process, we use introspection dynamics[38,39]. According to this process, players regularly compare their current payoff with the payoff they could have obtained by using a (randomly sampled) alternative strategy. The higher the payoff of the alternative, the more likely players are to switch (as described in more detail in the "Methods" section). If we apply this learning process to the three treatments, and if we artificially require players in the last two treatments to treat each repeated game as independent, all treatments yield equivalent results (Fig. S1). In particular, all treatments recover the qualitative findings of the previous literature on direct reciprocity[40]. In the following, we systematically explore the effect of linkage, by no longer imposing that players treat each game as independent.

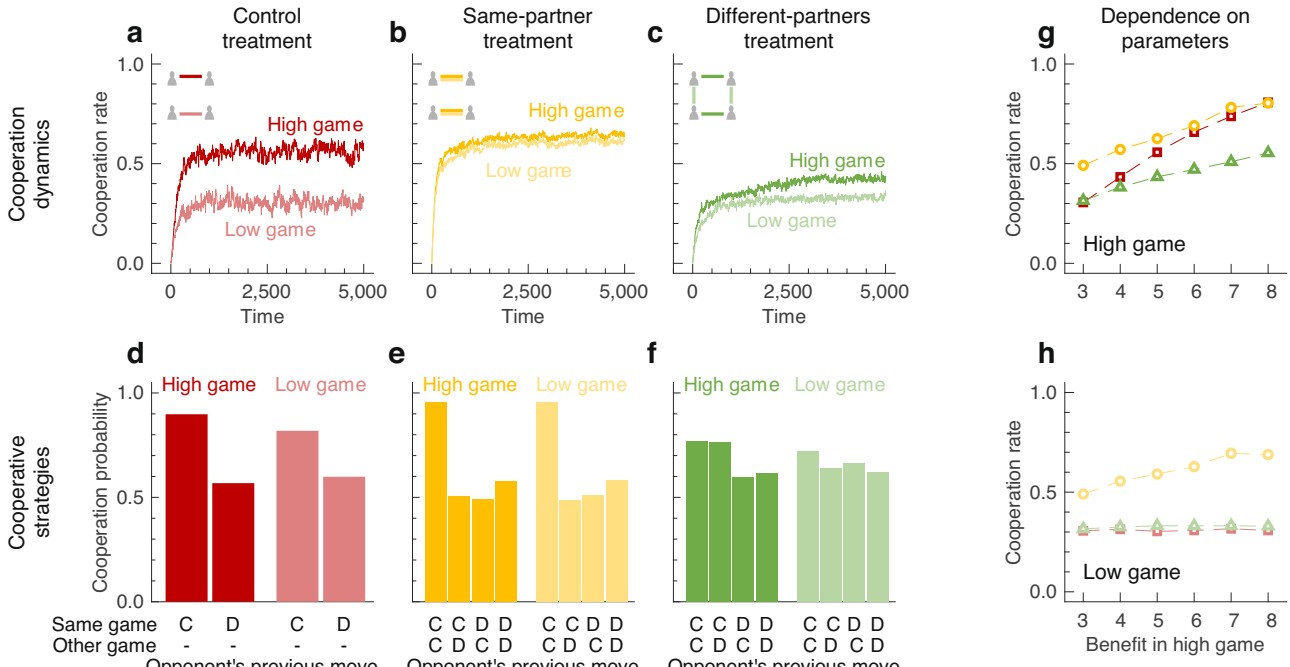

**Fig. 2 | Dynamics of cooperation in concurrent games. a–c** We use introspection dynamics[38,39] to model how people learn to cooperate in the three treatments. Here, we show average cooperation rates for both the high and the low game over time, averaged over 100 simulations. In the control treatment, there is substantially more cooperation in the high game than in the low game, as expected. In the same-partner treatment, players are generally more cooperative, whereas in the different-partners treatment, players tend to cooperate less. **d–f** We have recorded which strategies players use when they cooperate in both games (at least 2/3 cooperation rate). In the control treatment (red), players adopt strategies consistent with Generous Tit-for-Tat[9,10]. In the same-partner treatment (yellow), they only cooperate if the co-player previously cooperated in both games. In the different-partner treatment (green), individuals are still most cooperative after receiving cooperation in both games, but there is overall less cooperation (and very little linkage). C and D represent the decision to cooperate or to defect, respectively. **g** and **h** Our qualitative results remain valid for a wide range of parameter values (see also Fig. S3).

## Introspection dynamics of concurrent games

To get a first impression, we simulate the learning dynamics in the three treatments for fixed parameter values (in particular, we set $b_H = 5$, $b_L = 3$ and $c = 1$). Results in the control treatment recover the conventional wisdom established by previous work in direct reciprocity[40]. Repetition allows players to achieve some cooperation, and players are more cooperative when there is a high benefit (Fig. 2a). These intuitive results differ from what we find in both other treatments. In the same-partner treatment, individuals frequently cooperate in both games (Fig. 2b). These results confirm work by Donahue et al.[25] where players adopt strategies based on social comparisons rather than by introspection. In contrast, in the different-partners treatment, cooperation rates are consistently low (Fig. 2c). Because all three treatments yield equivalent results if players are artificially restricted to treat each game as independent (Fig. S1), these results indicate that linkage affects the cooperation dynamics. This effect is predicted to be favorable in the same-partner treatment, whereas it is detrimental when people play their games with different partners.

To explore the magnitude of linkage, we record the players' strategies during the learning process. In Fig. 2d–f, we report results for cooperative players (those with a cooperation rate of at least 2/3 in each game they participate in). In the control treatment, such players use strategies similar to Generous Tit-for-Tat, as one may expect based on the previous literature[9,10]. They tend to fully reciprocate a co-player's cooperation, and they show some leniency with defecting co-players (Fig. 2d). In contrast, evolving behaviors in the same-partner treatment are more strict. Here, players only fully cooperate in either game if the co-player previously cooperated in both games (they still show some leniency with respect to partial or full defectors, Fig. 2e). Importantly, these strategies exhibit linkage. Players condition their behavior in one game on actions that occurred in the other. The emergence of such strategies can explain why we see almost equal cooperation rates in both the high-benefit and the low-benefit game (Fig. 2b), although the incentives to cooperate in each game differ. Further simulations suggest that such strategies evolve in the same-partner treatment because they are more stable, compared to a strategy that just uses Generous Tit-for-Tat in each game (Fig. S2). Finally, for the different-partners treatment, players are unlikely to cooperate in both games altogether. Even when both co-players cooperated in the previous round in their respective games, players are, on average, less likely to reciprocate, and there is also little linkage overall (Fig. 2f).

## Robustness beyond the donation game

Overall, the same-partner treatment results in more cooperation, whereas the different-partners treatment leads to reduced cooperation rates. These qualitative findings are robust. In particular, they do not depend on the exact benefit of cooperation (see Fig. 2g, h, which shows results for different values of $b_H$). Similarly, they neither depend on whether or not players commit implementation errors (Fig. S3a, b); nor do they depend on whether or not the game is infinitely repeated (Fig. S3c, d). Finally, the qualitative results do not change as we vary the strength of the selection parameter, which describes the efficiency of the introspection learning process (Fig. S3e, f).

Moreover, in the Supporting Information, we show that the framework can be further extended to describe scenarios where the stage games are different from the donation game. For example, in Fig. S4, we present simulation results when the low-benefit game is replaced by a (general) prisoner's dilemma, a snowdrift game, or a coordination game. In each case, we find considerable differences between the three treatments. Except for the case of a coordination game, we also find that the same-partner treatment tends to generate the largest payoffs.

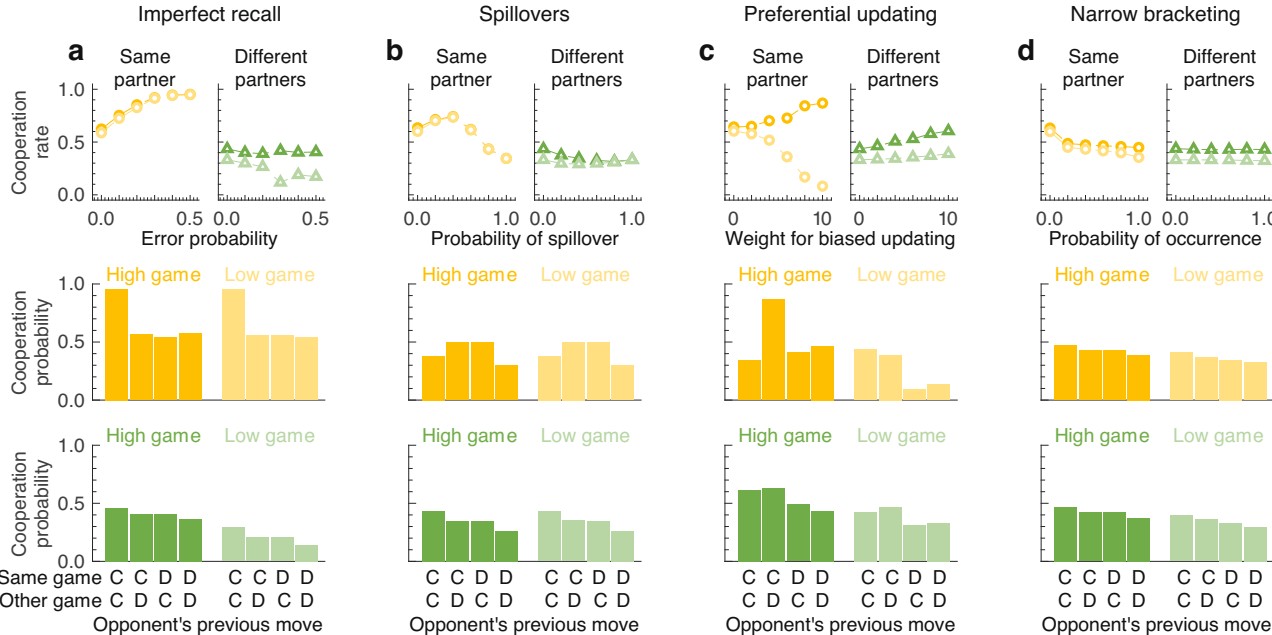

**Fig. 3 | Modeling the effect of cognitive constraints in concurrent games.** Our framework allows us to study the effect of various constraints, biases, and heuristics on cooperation in concurrent games. Here, we explore the impact of **a** imperfect recall, **b** spillovers, **c** preferential updating in the game with lower payoffs, and **d** narrow bracketing. In each case, we record the impact on average cooperation rates (upper panel). In addition, we also record the evolving average strategies in the most extreme case (lower two panels). C and D represent the decision to cooperate or to defect, respectively. For the same-partner treatment (yellow), we find that spillovers and narrow bracketing are most detrimental to cooperation. In that case, average cooperation rates may even be below the cooperation rates of the baseline control treatment (Fig. 2a).

We conclude that linkage in concurrent games has substantial effects on cooperation, across a wide range of reciprocal relationships.

## Incorporating cognitive constraints and different learning heuristics

Our framework allows us to go beyond a mere comparison between concurrent games and classical models of direct reciprocity. Instead, we can also explore the consequences of several cognitive constraints that are impossible to study (or have no analog) in classical single-repeated games. In the following, we introduce four model variations. Each model variation discusses a different type of constraint or heuristics that may conceivably affect behavior in concurrent games. In each case, we briefly summarize how they can be incorporated into our framework and how they affect our results. For all derivations and a more detailed discussion, we refer to the Supporting Information.

The first model extension addresses the impact of *imperfect recall*. Everyday experience and previous experiments[29–31] suggest that people with several interactions may confuse past outcomes. A co-player's cooperation in one game may be misremembered as having happened in a different game, possibly with a different co-player. To capture this form of imperfect recall across games, we assume players confuse past outcomes with probability $\varepsilon_{IR} \geq 0$. When such an error occurs, instead of correctly recollecting the previous actions in the high and the low game as $(a^H, a^L) \in \{C, D\}^2$, the player takes them to be $(a^L, a^H) \in \{C, D\}^2$. As a result, the player cooperates with probability $p^k_{a^L, a^H}$ instead of $p^k_{a^H, a^L}$. This type of error differs from simpler types of confusion or assessment errors[41], where players merely mislabel a co-player's past action with a fixed probability. In particular, errors of this kind have no effect if the previous outcome is either (C, C) or (D, D), or if the player's strategy happens to satisfy $p^k_{CD} = p^k_{DC}$. In the first case, no confusion between the two games can arise, whereas in the second case, any confusion proves to be inconsequential. Errors of imperfect recall can arise both in the same-partner treatment and the different-partners treatment. Yet they may have more of an effect when

interacting with different partners, as they might lead players to give misdirected responses[30,31]. In line with this intuition, we find that such errors have a weakly negative effect on cooperation in the different-partners treatment (Fig. 3a). Perhaps surprisingly, however, we find that imperfect recall reinforces cooperation in the same-partner treatment. Here, errors provide further incentives for players to link their behavior across games, and to only cooperate if the co-player previously cooperated in both games (Fig. 3a).

The second model extension addresses (exogenous) *behavioral spillovers*. A spillover arises when an individual's action in one domain leads that individual to take the same action in a different domain. Such spillovers have been reported in various contexts, and they can have important policy implications[42,43]. In our context, spillovers introduce additional correlations into a player's behavior. For any given history, they increase the chance that a player chooses the same action in each of the two games (rather than cooperating in one game and defecting in the other). For the same reason as before, such correlations seem particularly harmful when interacting with different partners because they undermine a player's ability to give targeted responses. Indeed, simulations again suggest a weakly negative effect of spillovers in the different-partners treatment (Fig. 3b). In contrast, in the same-partners treatment, the effect can be both positive and negative, depending on how frequent spillovers are. Indeed, in some cases, the resulting cooperation rates may even be below the cooperation rates of the control treatment (Fig. 2a).

The next two model extensions address different ways how people might update their strategies in the two games. In our previous simulations, we assume that players are equally likely to update their strategy in either the high or the low game. Instead, players may be more inclined to update their strategy in the game in which they currently receive the smaller payoff (relative to the maximum feasible payoff in that game). Simulations suggest that such *preferential updating* has weakly positive effects on the different-partners treatment. In the same-partner treatment, it increases cooperation in the

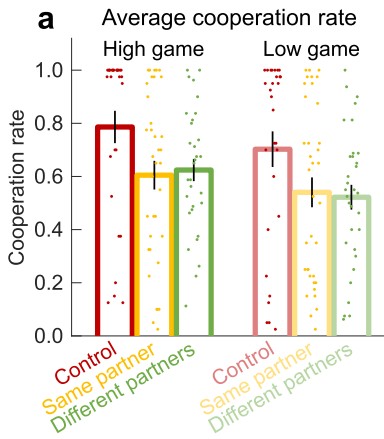

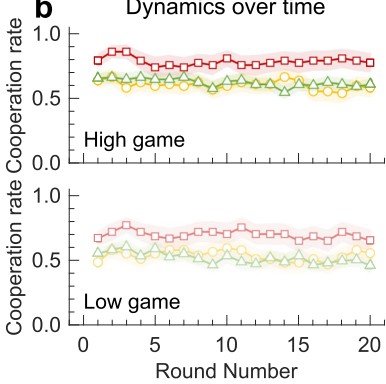

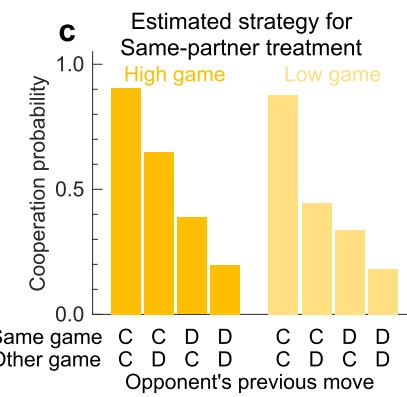

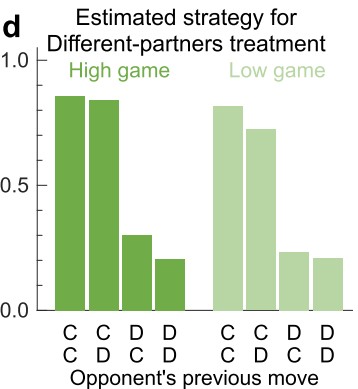

**Fig. 4 | Concurrent games among humans.** To explore how people act in concurrent games, we have implemented a behavioral experiment using the three treatments in Fig. 1. Participants are randomly matched and interact for at least 20 rounds. After that, the game continues with a 50% probability each round. **a** The vertical axis refers to the averaged cooperation rates of each group of two or four players ($n = 97$) over all 20 first rounds, for each treatment and across both the high and the low games (horizontal axis). Across all rounds, people were most cooperative in the control treatment (red), in both the high and the low game. Cooperation rates in the same-partner (yellow) and the different-partner treatment (green) are not significantly different from each other. Data are presented as mean values, individual data points are overlaid as dots and the SEM is represented by a vertical bar. **b**, These qualitative results are already present in the first round, and they are stable throughout the experiment. Data are presented as mean values with shaded areas representing the SEM. **c, d** We use linear regression to estimate the players' strategies based on the co-player's behavior in the previous round (Table 1). Here, we visualize the resulting conditional cooperation probabilities. C and D represent the decision to cooperate or to defect, respectively. In the same-partner treatment, participants link their behavior across games. As a result, a player's cooperation probability depends on the previous outcome of both games. In comparison, behaviors in the different-partners treatment are largely independent across the two games.

high game, but it destabilizes cooperation in the low game (Fig. 3c), presumably because players now update their low-game strategies more often.

Our last model extension addresses *narrow bracketing*. Narrow bracketing refers to situations in which people make decisions in one domain, without fully internalizing the consequences of those decisions in a different domain[44]. Such a bias may also affect how people learn in concurrent games. When players update their strategies in one game (high or low), they may not anticipate how these changes affect the dynamics of the other game. Narrow-bracketing has limited effects when people naturally treat their games as independent. In that case, changes in one game's strategy have no effect on the dynamics of the other. As a result, simulations suggest that narrow bracketing has no discernible impact in the different-partners treatment (Fig. 3d). However, in the same-partner treatment, in which players naturally learn to link their behavior across games, the effects can be considerable. Here, we find that narrow-bracketing undermines cooperation, both in the high and the low game.

Overall, our framework can readily capture each of the four cognitive constraints and learning heuristics discussed above. While all of them seem to have practical relevance, they have been rarely discussed in the context of direct reciprocity. Here, we have shown how each of these constraints and heuristics can be easily formalized within the context of concurrent games.

## Human behavior in concurrent games

The previous theoretical results indicate that concurrent games can alter the dynamics of reciprocal interactions. But whether concurrent games lead to more or less cooperation depends on how they update their strategies, whether their decision-making is influenced by biases and heuristics, and whether they interact with the same or with different partners. To explore the actual cooperation dynamics among human participants in more detail, we conducted a behavioral experiment. The experiment directly implements the three treatments illustrated in Fig. 1b–d. Participants are randomly assigned to treatments, and in the control treatment, they are randomly assigned to either play the high or the low game. In the high game, players can pay 2 points to give 4 points to the other player. In the low game, a player's 2 points are translated into 3 points for the co-player. Participants interact for at least 20 rounds, with a stochastic stopping rule implemented thereafter.

Figure 4a shows the resulting average cooperation rates across three treatments. In contrast to the predictions of the baseline model, but in agreement with some of our model extensions, we find that

people are most cooperative in the control treatment. More specifically, the average cooperation rate in the high game is 78.6% in the control ($n = 29$), compared to 60.5% in the same-partner ($n = 36$) and 62.4% in the different-partners ($n = 32$) treatment ($p = 0.004$, $\delta = 0.39(95\% \text{CI}[0.10, 0.62])$ for same-partner, $p = 0.006$, $\delta = 0.43(95\% \text{CI}[0.11, 0.67])$ for different-partners). Similarly, cooperation rates in the low game are 70.3% in the control, compared to 54.0% for the same-partner treatment and 52.2% for the different-partners treatment. However, here only the difference in the same-partner treatment is significant ($p = 0.056$, $\delta = 0.28(95\% \text{CI}[-0.02, 0.53])$ same-partner, $p = 0.007$, $\delta = 0.40(95\% \text{CI}[0.08, 0.65])$ different-partners). Interestingly, these differences in cooperation rates are already present in the first round, and they are stable throughout the experiment (Fig. 4b). Analysis of the first-round data shows that this difference is significant only in the high game for the same-partner treatment $p = 0.021$, $\delta = 0.33(95\% \text{CI}[0.02, 0.57])$, and in the low game for the different-partner treatment $p = 0.024$, $\delta = 0.29(95\% \text{CI}[0.04, 0.51])$. These results suggest that the simultaneous presence of two games has its own effect on cooperation independently of reciprocal dynamics.

To further understand these results, we look at the underlying conditional cooperation rates. Theoretical results suggested that players use different underlying strategies depending on whether the games are played with the same or different partners, and that they link the two games only in the same partner treatment. To explore to which extent linkage is also present in the experimental data, we infer the participants' reactive strategies based on their actual decisions. For any possible outcome of the previous round, we estimate how likely participants are to cooperate in the next round, both for the high and the low game. The results are summarized in Fig. 4c and Table 1. In line with our earlier simulations, linkage is more pronounced in the same-partner treatment. For example, in the high game, a linear regression suggests that participants cooperate with a 90.5% probability if the co-player previously cooperated in both games. If the co-player only cooperated in the high game, this cooperation probability drops to 64.8%. In comparison, the linkage is much weaker in the different-partners treatment. For example, people cooperate with an 85.8% probability in the high game after receiving cooperation in both games. This number drops only marginally, to 84.2% when a player only received cooperation in the high game. More generally, Table 1 suggests that linkage only has a minor effect in the different-partners treatment. Still, overall cooperation rates are below the control treatment because players generally have lower cooperation probabilities (see Table S1 for the regression results for the control treatment).

Overall, and in line with our theoretical results, we observe the strongest linkage effects in the same-partner treatment. However, we also find that participants do not benefit from this linkage. Instead of using it to better enforce cooperation, participants end up cooperating less often than participants in the control treatment, leading to similar cooperation rates to the different-partners treatment. As a consequence, concurrent games result in reduced cooperation rates, independent of whether people have their games with the same partner or with different partners.

## Discussion

People routinely engage in several social interactions at once[45]. They cooperate with their friends, their colleagues, and their families, possibly all at the same time. Moreover, with any given interaction partner, people often have several independent interactions in parallel. Colleagues might work on several projects concurrently, and whole nations routinely interact and negotiate over a wide array of different policies[46]. Despite this prevalence of concurrent games, the main paradigm for direct reciprocity is to study cooperation in (isolated) repeated games. Such an approach is justified (and from a computational perspective even preferred) when people treat each game as

**Table 1 | A linear regression to estimate the magnitude of linkage in human participants**

| | Dependent variable | | | |
|---|---|---|---|---|
| | Cooperation | | | |
| | Same-partner | | Different-partners | |
| | High game | Low game | High game | Low game |
| Partner's previous decision in the high game ($C_{H,t-1}$) | 0.450 | 0.157 | 0.637 | 0.023 |
| | (0.103) | (0.059) | (0.050) | (0.041) |
| | $p < 0.001$ | $p = 0.007$ | $p < 0.001$ | $p = 0.582$ |
| Partner's previous decision in the low game ($C_{L,t-1}$) | 0.189 | 0.264 | 0.095 | 0.513 |
| | (0.055) | (0.082) | (0.042) | (0.055) |
| | $p = 0.001$ | $p = 0.002$ | $p = 0.024$ | $p < 0.000$ |
| Interaction ($C_{H,t-1}) \times (C_{L,t-1}$) | 0.068 | 0.275 | −0.079 | 0.070 |
| | (0.105) | (0.090) | (0.050) | (0.058) |
| | $p = 0.517$ | $p = 0.003$ | $p = 0.114$ | $p = 0.226$ |
| Constant | 0.198 | 0.181 | 0.205 | 0.210 |
| | (0.044) | (0.033) | (0.040) | (0.040) |
| | $p < 0.001$ | $p < 0.001$ | $p < 0.001$ | $p < 0.001$ |
| Observations | 1368 | 1368 | 2432 | 2432 |
| $R^2$ | 0.401 | 0.395 | 0.375 | 0.327 |
| Adjusted $R^2$ | 0.400 | 0.394 | 0.374 | 0.326 |
| Residual std. error | 0.379 (df = 1364) | 0.388 (df = 1364) | 0.384 (df = 2428) | 0.410 (df = 2428) |

Based on the data of our behavioral experiment, we estimate how likely participants are to cooperate, depending on their partner's previous behavior. In total, we have run six regressions (three treatments, in which two games are played each round). If participants treat each game as independent, we would expect that only the constant term and the partner's previous decision in the respective game affect a player's cooperation probability. However, in the same-partners treatment, we observe that also previous decisions in the other game have a significant impact (in the high game: $p < 0.001$, in the low game: $p = 0.007$). In addition, in the low game we observe a significant interaction of cooperation in the two games. In the different-partners treatment, we only observe a weak impact of the low game on high-game decisions. All other indicators for linkage are insignificant.

independent. However, herein we present a theoretical framework and experimental data that cast serious doubt on that assumption of independence.

For our theoretical analysis, we compare three idealized scenarios. In one scenario—the control—individuals only engage in one repeated game at a time, just as previously assumed by most of the literature. In the other two scenarios, individuals engage in two repeated games simultaneously, either with the same partner or with different partners. If individuals in these last two scenarios indeed were to treat each of their games as independent, all three scenarios yield indistinguishable results (Fig. S1). Yet as individuals learn to adopt more profitable strategies over time in an evolutionary process, we often find that they learn to link their behavior across games. This linkage is particularly pronounced when the different games take place among the same partners (Fig. 2), in which linkage can come with explicit strategic benefits[25–28].

By shifting the perspective from individual to interconnected games, our framework serves as a starting point to better describe the effects of different cognitive constraints and biases (Fig. 3). The previous literature on direct reciprocity focuses on implementation errors, or 'trembling hands'[35]. Such errors occur when individuals intend to cooperate but fail to do so, perhaps because of a lack of attention or of

resources. Previous empirical research, however, has documented a plethora of other constraints that conceivably affect how humans cooperate. For example, the work of Stevens and colleagues[30,31] shows how imperfect recall can undermine a person's ability to give directed responses. Our work suggests that the effects of imperfect recall depend on the previous history of interactions. People are only susceptible to this kind of error when they have made conflicting experiences, with cooperation in one game and defection in another. Moreover, the precise effects of imperfect recall also depend on an individual's strategy. While some strategies are sensitive to false recollections, others remain completely unaffected. In addition to imperfect recall, our framework can also capture several other plausible constraints and heuristics, such as spillovers between games, preferential strategy updating, and of narrow-bracketing. In this way, our framework systematically increases the scope of models of direct reciprocity.

While these model extensions add further realism, they make predictions more complex. For example, cooperation rates in concurrent games may be higher or lower than in classical repeated games, depending on whether games take place among the same or with different partners, and depending on the constraints that affect individual play. To explore the impact of concurrent interactions on human behavior we conducted a behavioral experiment. In line with recent work by Laferrière and colleagues[34], we find that overall cooperation rates in the same-partner and different-partners treatments are surprisingly similar. However, both of these treatments result in less cooperation than the single game control (a treatment that Laferrière et al. do not consider). These empirical results do not fully match the theoretical results when it comes to overall cooperation, but they do support the underlying difference in strategies found between the two treatments: while participants in the same-partner treatment routinely learn to link their behavior across games, the linkage is comparably weak if games take place among different partners (Fig. 4, Table 1). The difference between theory and experiment may stem from human participants treating playing two games simultaneously as intrinsically different from playing only one game at a time, as this effect appears already in the first round (Fig. 4b). To explore these results further, it would be valuable to see how the effect scales when participants interact in three or more games concurrently, or experience different combinations of games. Similarly, exploring behavior when the payoffs in each round are allowed to fluctuate, using the framework of stochastic games, would be equally intriguing[47–49].

Importantly, our empirical results put some natural bounds on previously suggested mechanisms for cooperation. First, in contrast to previous models of games among the same partners, concurrent games do not seem to promote reciprocity[25–28]. They rather make cooperation more fragile. Second, in contrast to previous work on generalized reciprocity and community enforcement[36,37], people in the different-partners treatment do not seem to be prepared to exploit their network structure to promote cooperation in the group. Here, too, the effect of multi-game contact appears to be negative.

Overall, our results suggest that models of direct reciprocity based on (single) repeated games only provide an incomplete picture of the reciprocal interactions around us. In concurrent games, individual experiences in one game can affect future behaviors in another. Such linkage between games lead to a richer dynamics, but they also make the emergence of reciprocal altruism more complex.

## Methods
In the following, we briefly summarize our theoretical and experimental methods. All details can be found in the Supporting Information.

### Calculation of payoffs
For each treatment, we compute the players' payoffs by representing the game as a Markov chain. The possible states of this Markov chain are the possible outcomes of a given (repeated or concurrent) game. For example, in the control treatment, consider players 1 and 2, who interact in a repeated donation game with high benefits. The possible outcomes of a given round are the four possible realizations $\mathbf{a} = (a^1, a^2) \in \{C, D\}^2$. Given the players' strategies and given the action profile $\mathbf{a}$ of the previous round, we can compute the probability $m_{\mathbf{a}, \tilde{\mathbf{a}}}$ that players choose actions according to the profile $\tilde{\mathbf{a}} = (a^1, a^2) \in \{C, D\}^2$ in the next round for each $\tilde{\mathbf{a}} \in \{C, D\}^2$. By computing all the possible transition probabilities, we derive a $4 \times 4$ transition matrix $M = (m_{\mathbf{a}, \tilde{\mathbf{a}}})$ that captures the dynamics of the repeated game. The respective invariant distribution $\mathbf{v} = (v_{CC}, v_{CD}, v_{DC}, v_{DD})$ describes how often we are to observe each possible outcome $(a^1, a^2) \in \{C, D\}^2$ on average. Given this invariant distribution, payoffs are given by

$$\pi^1 = (v_{CC} + v_{DC})b_H - (v_{CC} + v_{CD})c.$$
$$\pi^2 = (v_{CC} + v_{CD})b_H - (v_{CC} + v_{DC})c. \tag{4}$$

In the other treatments, payoffs can be calculated similarly, even though they require more computation. In the same-partner treatment, the possible outcomes of an interaction between players 1 and 2 are now given by a 4-tuple $\mathbf{a} = (a^{1H}, a^{2H}, a^{1L}, a^{2L})$. Here, an entry $a^{ik} \in \{C, D\}$ represents player $i$'s action in game $k$. Because each entry can take one of two values, there are now 16 possible outcomes. Hence the corresponding transition matrix is of size $16 \times 16$.

In the different-partner treatment, all four players need to be considered simultaneously. Therefore, the current state is now represented by an 8-tuple $\mathbf{a} = (a^{1H}, a^{2H}, a^{3H}, a^{4H}, a^{1L}, a^{2L}, a^{3L}, a^{4L}) \in \{C, D\}^8$. It follows that the state space has $2^8 = 256$ elements. Hence, calculating the players' payoffs requires the invariant distribution of a $256 \times 256$ transition matrix. While these particular computations are easily manageable when individuals interact across two games, the computational complexity increases exponentially with the number of games concurrently played.

Throughout the main text, our model is based on the assumption that people use reactive strategies to make their decisions, and that each game is infinitely repeated. Neither of these assumptions is strictly required. In fact, the computational complexity of the model is unchanged if we assume players to use so-called memory-1 strategies instead[35]. In that case, a player's action does not only depend on the co-players' actions in the previous round but also on their own previous actions. Similarly, the computational complexity is unchanged if we assume games are repeated with a constant continuation probability $\delta$. Also, in that case, payoffs follow from computing the invariant distribution of a $4 \times 4$, $16 \times 16$, and $256 \times 256$ matrix. The respective algorithm for the case of the control treatment is described, for example, in Ichinose and Masuda[50]. In Fig. S3, we show simulation results for $\delta < 1$.

### Description of the learning process
For our theoretical analysis, we take an evolutionary approach. Players adapt their strategies over time, depending on their payoffs (which in turn depend on the strategies of the other players). To model this adaptation process we use introspection dynamics. Compared to other processes, such as pairwise imitation[51], introspection dynamics has computational advantages and it is easier to simulate[38,39]. Moreover, we consider it to be the more natural dynamic when one player's best strategy does not necessarily result in good payoffs for another player. Such a case could occur, for example, when players interact with a different set of co-players (as in our different-partners treatment). In such cases, an evolutionary process based on pairwise imitation appears to be less plausible.

In the following, we describe our learning dynamics in detail. Learning happens in discrete time steps. For a given treatment, we assume that at time $t = 0$, players defect unconditionally, $\mathbf{p}^i = (0, \ldots, 0)$ for all players $i$. At each subsequent time point $t$, the following

elementary updating procedure happens. First, one of the players, say player $j$, is chosen at random. This player is then given an opportunity to revise its current strategy $\mathbf{p}^j$ in one of the two games. In the control treatment, the revision occurs in the one game the player is involved in. In the other two treatments, this is done by randomly choosing one of the two games $k \in \{H, L\}$. In each case, we replace player $j$'s strategy for game $k$ with a random strategy sampled from a uniform distribution ($j$'s strategy for the other repeated game is left unchanged). The player then compares this alternative strategy $\tilde{\mathbf{p}}^j$ to the current strategy $\mathbf{p}^j$. To this end, let $\pi^j$ be the payoff the player obtained with the current strategy $\mathbf{p}^j$. Similarly, let $\tilde{\pi}^j$ denote the payoff player $j$ would have got in the previous interaction when adopting strategy $\tilde{\mathbf{p}}^j$ instead (keeping the strategies of the other players unchanged). Player $j$ switches to the new strategy with a probability given by the Fermi-function[52,53]

$$\rho = \frac{1}{1 + \exp\left[-\beta(\tilde{\pi}^j - \pi^j)\right]}. \qquad (5)$$

The parameter $\beta \geq 0$ is the strength of selection. It measures to which extent strategy updates depend on payoffs. For $\beta \to 0$, payoffs are irrelevant and the updating probability approaches one half. In this limit of 'weak selection', updating occurs at random. In the other limit of 'strong selection', $\beta \to \infty$, only those alternative strategies are adopted that yield at least the payoff of the original strategy. Note that although this parameter $\beta$ also appears in other evolutionary models (e.g., in the birth–death processes or the pairwise comparison process), typical numerical values cannot be compared directly. For example, while several empirical studies estimate $\beta$ to be below one for these classical evolutionary processes[54–56], the same value of $\beta$ would induce a weak-selection regime for introspection dynamics (see Fig. S3e, f). Thus, for the simulations shown herein, we use a value of $\beta = 200$. However, the relative ranking of the different treatments is robust with respect to different selection strength values (Fig. S3e, f).

We iterate this elementary updating procedure for many time steps. For any finite selection strength $\beta$, this generates an ergodic stochastic process. In particular, the players' average cooperation rates (over the course of the learning process) converge in time, and they are independent of the players' initial strategies. For our study, we use simulations to numerically estimate these average rates for all three treatments.

We note that according to our implementation of the evolutionary process, individuals update their strategy either in the high-benefit or in the low-benefit game (but not in both). This implies that once all players adopt a mutually cooperative strategy, any mutant strategy would only deviate in one of the two games. This observation might explain why the strategies depicted in Fig. 2e are slightly more cooperative after a 'DD' outcome, compared to a 'CD' or 'DC' outcome. Because 'DD' outcomes occur less often, they are under weaker selection. However, we note that overall, our results are independent of the exact evolutionary process we use. In particular, for the same-partner treatment, similar results have been reported by Donahue et al.[25], based on a pairwise comparison process.

### Computational methods used for the figures

For the simulations in the main text, we use the following default parameters. The benefits in the two games are $b_H = 5$ and $b_L = 3$, respectively, and the cost is $c = 1$. In addition, we neglect any trembling-hand (implementation) errors, $\varepsilon_{TH} = 0$.

In Fig. 2a–c we show average trajectories. To this end, we have run 100 independent simulations for each treatment. To keep timescales comparable, simulations are run for 20,000 elementary time steps in the control treatment, and for 40,000 elementary time steps in the other two treatments. This implies that there are 5000 updating events per player and game on average in each treatment. In Fig. 2d and e, we

display which strategies cooperative players tend to use. To this end, we use the data of the simulations in Fig. 2a–c. We define a player's strategy to be cooperative if the player's cooperation rate in each game is at least 2/3 against the given co-player (other cut-offs give similar results). The panel then shows the arithmetic mean of all strategies classified as cooperative. Finally, Fig. 2g, h shows the impact of the benefit of cooperation in the high game. Here, each point corresponds to the time average of one long simulation ($10^6$ time steps). We explore the impact of other model parameters in Fig. S3.

In Fig. 3, we explore four model extensions that describe the impact of different cognitive constraints and heuristics. The top row in Fig. 3 describes how the evolving cooperation rates are affected as we change the probability $\varepsilon_{IR}$ of experiencing imperfect recall (Fig. 3a), the probability $\varepsilon_{SP}$ of experiencing a spillover (Fig. 3b), the weight $\kappa$ that measures the strength of preferential updating (Fig. 3c), and the likelihood $\lambda$ that a player engages in narrow bracketing (Fig. 3d). Each data point is the average of a simulation run for $10^6$ updating steps. The middle and the bottom row of the figure show the player's average strategies. This figure is based on all strategies used during the simulation (not only the cooperative strategies). For a more detailed discussion of each model extension, see Supporting Information.

### Experimental methods

For our experiment, we recruited 316 participants (161 females, mean age: 21) from the University of Exeter student pool: Finance and Economics Experimental Laboratory at Exeter (FEELE). All participants gave their informed consent, and we complied with all relevant ethical regulations. The experiment was approved by the Ethics Committee of the Medical Faculty of Kiel University (D 571/20) and is covered under the ethics approval number eUEBS001862 from the University of Exeter. The experiment was implemented in oTree[57]. Participants were matched in groups of four, all playing the same treatment. All participants were anonymous and only referred to by numbers from 1 to 4. Sessions were programmed for one of the three treatments and players only participated in one session. Each treatment lasted for a minimum of 20 rounds of the repeated game(s). After the 20th round, each subsequent round had a 50% chance of occurring, to avoid end-game effects. Participants received £3 for participating and could earn a bonus payment based on their decisions in the game. The points earned during the game were converted at a rate of 20 points = £0.26. The average bonus payment across all treatments was £1.39. In the same-partner and different-partners treatment, participants make their decision for both repeated games simultaneously, round by round. In the baseline treatment, they only take part in one repeated game. Payoffs in each repeated game are based on the payoff matrix of the donation game, with $b = 4$ points, $c = 2$ points in the high game, $b = 3$ points, and $c = 2$ points in the low game. These values were chosen based on preliminary simulations and pilots to avoid ceiling effects and to obtain the largest difference in cooperation rates between the two games in the control treatment.

We analyzed the data using two-tailed non-parametric tests as well as statistical regressions, using interacting pairs as statistical units, with the exception of the different-partner treatment where groups of four interacting participants are used (due to the nature of the design, these groups cannot be separated into pairs). This gives us 36 groups of 2 for the same-partner treatment, 32 groups of 4 for the different partners treatment, and 29 groups of 4 for the baseline control. The sample size was estimated from past research[34]. Four participants dropped out during the repeated game, two in the same-partner treatment and two in the control treatment. We calculated the average value for each pair/grouping of players, and we then compared this average value between treatments with a Mann–Whitney $U$-test, or within each treatment with a Wilcoxon signed-rank test. We report the outcome uncorrected for multiple tests, as all our main conclusions remain valid after correction. We only use the first 20 rounds of each

game for our analysis and only take into account groups without a drop-out. For more details on the experimental setup (see Supporting Information).

As a final note, experimental data such as ours can sometimes be used to estimate how people update their strategies[54–56]. In particular, one may attempt to estimate relevant parameters, such as the selection strength $\beta$. Unfortunately, our experimental design does not allow for this type of inference. According to our design, each participant only interacts in a single concurrent game (with fixed opponents). As a result, we cannot distinguish between individuals who use the same conditional (but fixed) strategy throughout the experiment and individuals who update their strategy midway[58]. To estimate how players update their strategies, we would need a different type of design. For example, future work could look at a setup where individuals play several consecutive concurrent games against changing opponents. Using such a design, a player's behavioral changes across games can be more readily interpreted as an instance of 'learning'.

### Reporting summary
Further information on research design is available in the Nature Portfolio Reporting Summary linked to this article.

## Data availability
Both the simulation data and the experimental data (raw) are available online under the https://doi.org/10.17605/OSF.IO/XQGHF.

## Code availability
All numerical simulations were performed with Matlab. The behavioral experiment was implemented with oTree[57], and the respective data was analyzed with R. The respective code is available online under the https://doi.org/10.17605/OSF.IO/XQGHF.

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

## Acknowledgements
C.H. acknowledges generous funding from the European Research Council (ERC) under the European Union's Horizon 2020 research and innovation program (Starting Grant 850529: E-DIRECT) and from the Max Planck Society. O.P.H. is grateful for generous funding from the University of Exeter Business School.

## Author contributions
C.S.L.R., O.P.H., and C.H. designed the research, performed the research, and wrote the paper concurrently.

## Funding

## Competing interests
The authors declare no competing interests.
