## [Transparent Peer Review file · Nature Communications]

Dynamics of cooperation in concurrent games

Corresponding Author: Ms Charlotte Rossetti

Version 0:

Reviewer comments:

Reviewer #1

(Remarks to the Author)

In this paper, authors study evolutionary dynamics of cooperation in concurrent repeated games. They use both computational modelling and simulations (game theory) and behavioural experiments in this study.

The paper addresses a timely and important problem, contributing to the literature of direct reciprocity and human prosocial behaviour. Although there have been many papers, both theoretical and empirical ones, that study how direct reciprocity can promote evolution of cooperation, there is still a crucial gap in exploring more realistic settings such as the one studied in this paper. That is, most previous works consider independent repeated interactions, unable to capture the fact that people usually engage in multiple dependent interactions where what's going on in one interaction can influence that in another one. To my knowledge, this is the first effort to systematically, formally as well as empirically address this important issue.

In general, the paper is well written. The theoretical models and the experimental design are well thought out. The analyses are done in a highly competent manner (including extensive additional results in supporting information).

Therefore, I am in favour of publication of the paper in Nature Communications, with only a few minor suggestions authors might want to consider:

1) The modelling analysis can be more thorough, considering the impact of varying the key parameters, such as b_H and b_L and the intensity of selection. For example, the motivation for choosing $\beta = 200$ in the simulations is not clear. This is a particularly strong intensity of selection, and previous behavioural experiments with human participants usually point to a weaker selection value (around 0.01 to 0.1) that best fits experimental data, see e.g. "Evolution of fairness in the one-shot anonymous ultimatum game." *Proceedings of the National Academy of Sciences* 110.7 (2013): 2581-2586. and "Generosity motivated by acceptance-evolutionary analysis of an anticipation game." *Scientific reports* 5.1 (2015): 18076.

It would be useful to check how the results change for varying this key parameter and discuss which one leads to behavioural observation that is most close to the experimental observation.

2) The paper focuses on introspection dynamics for modelling the behavioural update/learning process. This choice seems reasonable given that the two payoff matrices of the games can be different. However, in the experiments and the modelling analyses in the paper, the two games considered are both repeated Prisoner's dilemmas. I am wondering what's the actual behavioural update mechanism people in the experiments actually use, e.g. whether it's standard social learning (as both games are of the same strategic nature) or introspection. Also, what would happen if the two games are of different strategic nature, e.g. (anti-)coordination game vs prisoner's dilemma? It would be useful to at least discuss these.

(Remarks on code availability)

Reviewer #2

(Remarks to the Author)

The manuscript presents an insightful investigation into the complexities of cooperative behavior when individuals are engaged in multiple social dilemmas simultaneously. The authors introduce a theoretical framework to explore the crossover and spill-over effects in concurrent games, where the outcomes of one interaction can influence behavior in another. Employing an evolutionary learning process, the study reveals that individuals often link their behavior across games, a phenomenon that is particularly pronounced under cognitive constraints such as imperfect recall or narrow-bracketing. The manuscript also includes a behavioral experiment that complements the theoretical findings, suggesting that the interplay of concurrent games tends to negatively affect cooperation. This research significantly contributes to the understanding of strategic motives and spillovers in concurrent games, offering new perspectives on reciprocity and cooperation dynamics.

The manuscript is well-prepared and contributes meaningfully to the existing literature. It has laid a solid foundation for future research in the field of cooperative game dynamics.

While the manuscript is of high quality, there are several questions that could be posed to the authors for further consideration and potential refinement of their work:

1) In Figure 2b, the cooperation rates in the high-benefit game and the low-benefit game appear to be unexpectedly similar. What factors within your model contribute to this observation, despite the intuitive expectation that the cooperation rate in the low-benefit game should be lower?

2) In Figure 2e, it is noted that the cooperation probability in both the high and low games exceeds 50% when an opponent selects strategy D in the same game as well as in the other game. This rate is unexpectedly higher than when the opponent chooses C in the same game and D in the other (or D in the same and C in the other). What mechanisms within your model could account for this counterintuitive finding? Could the authors elucidate the factors that might drive individuals to cooperate at a higher rate despite facing defection in both games?

3) Does the game environment remain constant throughout the repeated game interactions, or is there potential for it to evolve, possibly impacting the strategic choices of the players? Specifically, does the environment ever shift from consistently being a 'high game' or a 'low game' to something else, and if so, how does this affect cooperation levels? In light of the findings in Nature 559, 7713 (2018): 246-249, which discuss the evolution of cooperation under different environmental conditions, could the authors explore whether similar principles apply to the transitions between 'high' and 'low' game contexts in your study?

4) In the memory effect model presented in your study, imperfect recall is described as occurring between the high game and the low game contexts. What cognitive or theoretical rationale underpins this specific form of recall error? Is it due to the distinct nature of the games or the way decisions are framed within each game? Could the authors elaborate on why imperfect recall might manifest as confusion between games rather than as misremembering the specific strategies of an opponent?

5) In the context of concurrent repeated games, your study assumes that outcomes in one game can influence decision-making in another, a phenomenon that could be likened to 'crosstalk.' How does the mechanism of decision linkage you've identified differ from the 'crosstalk' effect described by Reiter, Johannes G., et al. in 'Crosstalk in concurrent repeated games impedes direct reciprocity and requires stronger levels of forgiveness,' published in Nature Communications 9.1 (2018): 555? Could the authors provide a comparative analysis of these two concepts and discuss how they might interact or diverge within the framework of concurrent games?

(Remarks on code availability)

Reviewer #3

(Remarks to the Author)

How to understand the evolution and emergence of cooperation has attracted widespread attention from a broad range of disciplines in recent years. Direct reciprocity is one of the core mechanisms enabling cooperation among unrelated individuals. Yet most of the works regarding direct reciprocity assume that individuals either only engage in one repeated game at a time or that they treat each game as independent. Indeed, in most real examples, individuals engage in several games concurrently. In concurrent games, the outcome of one interaction may affect how individuals subsequently behave in a different game. In this manuscript, the authors propose a theoretical framework of concurrent games. They assume that individuals repeatedly engage in two independent stage games, either with the same or with different partners and agents adapt the strategies over time according to an evolutionary learning process. The authors find that individuals often link their behaviors across the two games and the linkage is particularly pronounced when the different games take place among the same partners. They also study four constraints and heuristics that may conceivably affect behavior in concurrent games. The authors show that the linkage in the concurrent games can have both a positive or a negative effect on cooperation. Additionally, they conduct a behavioral experiment to further explore the theoretical results. However, their experimental results show that the overall effect of two concurrent games on cooperation tends to be negative.

I have enjoyed reading this paper. I have found it well-written and well-organized and the results are interesting. It is my opinion that this manuscript systematically investigates the dynamics of cooperation in concurrent games and makes a contribution to the field. However, I also have some following comments or questions on this work.

In this work, the authors only consider that individuals repeatedly engage in two independent stage games in the framework of concurrent games they proposed. Indeed, individuals engage in two or more games simultaneously in concurrent games. From this point, the authors can propose a more general framework for concurrent games. In addition, they can also show whether their results presented in this paper are still valid when individuals engage in more than two games simultaneously. Besides, in concurrent games individuals may engage in different types of social dilemma games, not only play the donation game, as shown in Fig. S4. The authors may have some more discussion about this point in the manuscript.

The authors mainly present the cooperation rates in high and low games, respectively. It is fine. However, when individuals engage in concurrent games, there exist the resulting cross-over and spill-over effects. Readers may be interested in such effects on the overall cooperation rate in concurrent games. From this point, the authors can present some corresponding results in the manuscript.

The authors explore theoretically and experimentally how different strategic motives and spillovers jointly shape individual actions on concurrent games. Their theoretical results show that the linkage in concurrent games can have both a positive or a negative effect on cooperation, compared to the standard case of independent games. However, their behavioral experiment suggests the overall effect of two concurrent games on cooperation tends to be negative. I think that it is important to explain why such discrepancy of theoretical predictions and experimental results can happen.

I note that the idea of concurrent games has been proposed in previous works (see Ref. 25 and Ref. 26). Meanwhile, Ref. 26 shows that the crosstalk effect in concurrent games impedes the maintenance of cooperation. From this point, the authors should point out the focus of this present study and importantly they should stress the differences between their work and Refs. 25 and 26. In this present version of the manuscript, however, the authors did not make a deeper comparison. The novelty and contribution of this work need to be justified.

(Remarks on code availability)

Reviewer #4

(Remarks to the Author)

This work titled "Dynamics of cooperation in concurrent games," extends the traditional evolutionary game theory by developing an innovative theoretical framework to examine cooperation dynamics in environments where individuals participate in multiple concurrent games. Through a synthesis of theoretical modeling and behavioral experiments, the research delves into the intricacies of direct reciprocity, transcending the conventional presupposition of treating games in isolation.

Employing introspection dynamics, the authors simulate the strategic evolution of cooperation within a controlled experimental design that accounts for cognitive limitations and heuristic-driven learning processes. The investigation uncovers that cross-game linkages can exert a bidirectional influence on cooperative behaviors, potentially enhancing or undermining the propensity for collaboration.

The findings reveal a nuanced relationship between concurrent game engagement and cooperative outcomes, with significant implications for economic, psychological, and sociological perspectives on human interaction. This work advances our grasp of the factors that catalyze or impede the emergence of cooperative strategies in complex social settings.

I have taken note of the authors' investigation into the dynamics of cooperation in concurrent games, especially their findings regarding how individual behaviors can influence each other across different games and form cross-game linkages. These findings resonate with the conclusions of the authors' previous work "Evolution of cooperation in stochastic games" (Nature 559, 7713 (2018): 246-249).

However, to ensure the novelty and substantial contribution of this paper to the existing literature, I recommend that the authors:

Clearly delineate the differences and connections between this work and previous studies in terms of theoretical framework, experimental design, or data analysis.

Elaborate on the innovative aspects of this study compared to previous work, especially the new cognitive constraints and learning heuristics factors introduced when considering concurrent games.

Discuss how this study further validates or extends previous theoretical models through experimental methods, and what new insights these experimental results provide for understanding cooperative behavior in concurrent games.

(Remarks on code availability)

Version 1:

Reviewer comments:

Reviewer #1

(Remarks to the Author)

The authors have addressed all my previous comments very well. I really appreciate the extra efforts they put in to improve the paper in response to all the reviewers' comments. I am happy to recommend its publication in the present form.

(Remarks on code availability)

Reviewer #2

(Remarks to the Author)

The paper presents good findings and contributes insights to the field of EGT. The research methodology is robust, and the conclusions are well-supported by the data presented. Given that the previously raised issues have been satisfactorily addressed by the authors, I am now in full agreement with the publication of this manuscript in Nature Communications.

(Remarks on code availability)

Reviewer #3

(Remarks to the Author)

The authors have provided reasonable replies to my comments and revised the manuscript accordingly. I would like to recommend the publication of the work.

(Remarks on code availability)

First of all, we would like to thank the reviewers for their valuable feedback, and the editor for his invitation to submit a revised manuscript. We appreciate the many constructive comments that the reviewers have raised. In the meantime, we have addressed all the comments. In particular, we now describe more clearly how our paper relates to previous work, how some of our results need to be interpreted, and what happens as we vary different model parameters. We would like to thank the reviewers for instigating these changes. We believe they improved the paper considerably. Please find a detailed point-by-point reply below.

Reviewer #1

In this paper, authors study evolutionary dynamics of cooperation in concurrent repeated games. They use both computational modelling and simulations (game theory) and behavioural experiments in this study. The paper addresses a timely and important problem, contributing to the literature of direct reciprocity and human prosocial behaviour. Although there have been many papers, both theoretical and empirical ones, that study how direct reciprocity can promote evolution of cooperation, there is still a crucial gap in exploring more realistic settings such as the one studied in this paper. That is, most previous works consider independent repeated interactions, unable to capture the fact that people usually engage in multiple dependent interactions where what's going on in one interaction can influence that in another one. To my knowledge, this is the first effort to systematically, formally as well as empirically address this important issue.

In general, the paper is well written. The theoretical models and the experimental design are well thought out. The analyses are done in a highly competent manner (including extensive additional results in supporting information). Therefore, I am in favour of publication of the paper in Nature Communications, with only a few minor suggestions authors might want to consider:

Reply: Thank you for the positive feedback, and for the detailed comments below!

1. The modelling analysis can be more thorough, considering the impact of varying the key parameters, such as b_H and b_L and the intensity of selection. For example, the motivation for choosing $\beta = 200$ in the simulations is not clear. This is a particularly strong intensity of selection, and previous behavioural experiments with human participants usually point to a weaker selection value (around 0.01 to 0.1) that best fits experimental data, see e.g. "Evolution of fairness in the one-shot anonymous ultimatum game." *Proceedings of the National Academy of Sciences* 110.7 (2013): 2581-2586. and "Generosity motivated by acceptance-evolutionary analysis of an anticipation game." *Scientific reports* 5.1 (2015): 18076. It would be useful to check how the results change for varying this key parameter and discuss which one leads to behavioural observation that is most close to the experimental observation.

Reply: This is a very good point. Because many of our results are based on computer simulations, it is important to explore how robust our results are with respect to parameter variations. In fact, already in our earlier submission we have reported such results. For example, in **Figure 2g,h**, we systematically vary the high-game's benefit b_H . Similarly, in **Figure S3**, we explore the impact of several other parameters including the strength of selection. We find that while all these parameters have some impact on the evolving overall cooperation rate, they usually do not affect the qualitative ranking between the different treatments. However, the reviewer's comment made us realize that we hardly discussed these robustness checks in the main text. In the revised manuscript, we mention them more explicitly.

We also appreciate the comment regarding the strength of selection. As mentioned by the reviewer, we use $\beta = 200$ as our default value. Compared to the above cited papers, this value indeed seems to be rather large. However, please let us note that we use a different learning process, and hence the values of β cannot be compared directly. For example, in the paper by Rand et al on the ultimatum game, the authors use a birth-death process. In such processes, there are two variables that jointly control the amount of noise in the evolutionary process: the population size N and the selection strength β . As a rule of thumb, the larger the population size, and the larger the selection strength, the more deterministic the stochastic process becomes. In particular, for moderate population sizes such as $N = 100$, even a comparably small selection strength of $\beta = 0.1$ can considerably favor the selection of strategies with higher payoffs. In contrast to those papers, we use introspection dynamics as our evolutionary process. This process describes learning on the group-level, instead of the population-level. In particular, there is no population size parameter any more. For introspection dynamics, a selection strength of $\beta = 1$ in fact corresponds to ‘weak selection’. This aspect also becomes clear in our **Figure S3e,f**, where all evolving cooperation rates are close to the neutral value of $\frac{1}{2}$ when the selection strength is one. The same graph also shows that our value of $\beta = 200$ is somewhat intermediate. It is strong enough for payoff differences to have a notable effect. But at the same time, a further increase in β would still affect how much cooperation evolves across the three treatments.

Changes: In the revised manuscript we describe the effects of different parameter changes in more detail. Moreover, in the methods section, we describe why the selection strength parameter according to introspection dynamics cannot be directly compared to the analogous parameter in birth-death models (as in the papers cited above). Finally, we also give some justification why we used introspection dynamics to begin with. In short, introspection dynamics makes some computations more efficient (see e.g., Couto et al, *New Journal of Physics*, 2022). In addition, we regard it to be a more sensible process to describe learning when individuals do not interact with an entire population, but only with their given neighbors.

2. The paper focuses on introspection dynamics for modelling the behavioural update/learning process. This choice seems reasonable given that the two payoff matrices of the games can be different. However, in the experiments and the modelling analyses in the paper, the two games considered are both repeated Prisoner’s dilemmas. I am wondering what’s the actually behavioural update mechanism people in the experiments actually use, e.g. whether it’s standard social learning (as both games are of the same strategic nature) or introspection.

Reply: That is a great question. Indeed, it would be interesting to explore which kind of learning algorithm people use. However, that question gives rise to an entire research project on its own, and our experiment was not designed to address that question. Strictly speaking, with an experiment such as ours, we cannot even tell whether individuals updated their strategies during the experiment at all.

To see why, consider a (single) repeated prisoner’s dilemma, and suppose two players interact for twenty rounds. Moreover, assume the first player defects in all twenty rounds, whereas the other player cooperates in the first round and then defects in all remaining rounds. There are two alternative explanations for the second player’s (hypothetical) behavior. The first explanation is that the second player used a conditionally cooperative strategy (like Tit-for-Tat) throughout the experiment. Under this interpretation, this player never updated her strategy. Rather, she just implemented the actions that her strategy prescribed, and she kept her strategy throughout the game. The other explanation is that this second player updated her strategy after the first round, possibly because she compared her payoffs to

the payoffs of her opponent. Our data cannot differentiate between these two explanations. Both are equally compatible with the observed behavior. To make a clear case for learning, one would rather need a design in which participants engage in a series of repeated games, against changing opponents. That design would be more suitable to detect learning across repeated games.

Changes: We now mention in the Methods section that with our data, we cannot infer the participants' updating mechanism. The corresponding paragraph follows the argument given above.

Also, what would happen if the two games are of different strategic nature, e.g. (anti-)coordination game vs prisoner's dilemma? It would be useful to at least discuss these.

Reply and Changes: Thanks for this comment, we fully agree! In fact, already in our previous submission we addressed this question (but again, the respective results were somewhat hidden in the SI). In our **Figure S4**, we explore different cases, depending on whether the low game is a donation game, a general prisoner's dilemma, a snowdrift game, or a coordination game. In our revised manuscript, we now mention these simulations more prominently within the main text.

Reviewer #2

The manuscript presents an insightful investigation into the complexities of cooperative behavior when individuals are engaged in multiple social dilemmas simultaneously. The authors introduce a theoretical framework to explore the crossover and spill-over effects in concurrent games, where the outcomes of one interaction can influence behavior in another. Employing an evolutionary learning process, the study reveals that individuals often link their behavior across games, a phenomenon that is particularly pronounced under cognitive constraints such as imperfect recall or narrow-bracketing. The manuscript also includes a behavioral experiment that complements the theoretical findings, suggesting that the interplay of concurrent games tends to negatively affect cooperation. This research significantly contributes to the understanding of strategic motives and spillovers in concurrent games, offering new perspectives on reciprocity and cooperation dynamics. The manuscript is well-prepared and contributes meaningfully to the existing literature. It has laid a solid foundation for future research in the field of cooperative game dynamics.

Reply: Thank you for this positive summary, and thank you for the many constructive suggestions below.

While the manuscript is of high quality, there are several questions that could be posed to the authors for further consideration and potential refinement of their work:

1. In Figure 2b, the cooperation rates in the high-benefit game and the low-benefit game appear to be unexpectedly similar. What factors within your model contribute to this observation, despite the intuitive expectation that the cooperation rate in the low-benefit game should be lower?

Reply: This part of our results replicates similar findings from Donahue et al (2020). They also found that when players simultaneously engage in several games against the same partner, cooperation in one game (typically the high-benefit game) can be used to enforce cooperation in the other. The intuition is the following. In the same-partner treatment, players can link their behavior across games. They can adopt a strategy to only cooperate in either game if the opponent previously cooperated in both games.

Once both players adopt such a strategy, they have an incentive to equally cooperate in the high-benefit and the low-benefit game. This intuition is also reflected in our simulation results. For example, **Figure S1** shows that once players in the same-partner treatment are exogenously prevented from linking their behavior across games, cooperation rates in the high-benefit game are higher than in the low-benefit game (as the reviewer expected).

Changes: We have revised our description of the results in **Figure 2b**, to provide a better intuition.

2. In Figure 2e, it is noted that the cooperation probability in both the high and low games exceeds 50% when an opponent selects strategy D in the same game as well as in the other game. This rate is unexpectedly higher than when the opponent chooses C in the same game and D in the other (or D in the same and C in the other). What mechanisms within your model could account for this counterintuitive finding? Could the authors elucidate the factors that might drive individuals to cooperate at a higher rate despite facing defection in both games

Reply: This is a very good point. Indeed, **Figure 2e** suggests that players are slightly more likely to cooperate if their opponent defected in both games (DD) than when this opponent only defected in one game (CD or DC). This is somewhat counterintuitive. Naively, we would expect that players who experienced a defection in both games would be the ones most likely to retaliate.

To interpret these observations, note that according to our learning process, individuals usually only update their strategies in one of the two games (see our Methods section). This means that once full cooperation is established, mutant strategies will typically not start defecting in both of the games, but only in one of the games. By having a small cooperation probability after CD and DC, players ensure that their strategy is stable against such mutants. Of course, they cannot afford being too generous after a DD either; but because such events are rarer, these events are less relevant for a player's payoff.

Changes: We now provide a similar interpretation in the methods section.

3. Does the game environment remain constant throughout the repeated game interactions, or is there potential for it to evolve, possibly impacting the strategic choices of the players? Specifically, does the environment ever shift from consistently being a 'high game' or a 'low game' to something else, and if so, how does this affect cooperation levels? In light of the findings in Nature 559, 7713 (2018): 246-249, which discuss the evolution of cooperation under different environmental conditions, could the authors explore whether similar principles apply to the transitions between 'high' and 'low' game contexts in your study?

Reply and Changes: Indeed, for this project, we consider a constant environment. This is one of the major differences compared to previous work on stochastic games, mentioned by the reviewer. However, looking at concurrent games in changing environments would be an interesting project. We now mention this possible direction for future work in our Discussion. Thank you for mentioning this idea!

4. In the memory effect model presented in your study, imperfect recall is described as occurring between the high game and the low game contexts. What cognitive or theoretical rationale underpins this specific form of recall error? Is it due to the distinct nature of the games or the way decisions are framed within each game? Could the authors elaborate on why imperfect recall might manifest as confusion between games rather than as misremembering the specific strategies of an opponent?

Reply: The reviewer is absolutely correct. In our study, when we mention “imperfect recall”, we refer to the following situation: when a player experiences cooperation in one game and defection in the other, this player may sometimes mistake the game in which the defection happened. Alternatively, one could also consider an alternative type of error: when a player experiences a cooperation in a given game, this player may remember it as a defection (independent of what happened in the other game). This type of mistake is sometimes called ‘observation error’ (because an observed cooperation is taken to be a defection, and vice versa, e.g., Schmid et al, *Nature Human Behaviour* 2021).

Our analysis of imperfect recall is motivated by previous studies by Stevens and colleagues (e.g., *Frontiers in Psychology* 2011). They found that human subjects find it difficult to accurately remember the previous actions of different co-players. Moreover, these difficulties become more pronounced the more co-players there are. In our opinion, these observations are more in line with an interpretation that individuals confuse their past interactions. For example, if all co-players of a participant cooperated in the past, we think it is unlikely that any participant would mistakenly believe that some co-player defected. This hypothetical example is consistent with our modeling of imperfect recall (errors can only happen if there are several conflicting experiences). The example is inconsistent with a model in which a player just misremembers an opponent’s past action with some constant probability.

Having said that, we think both types of errors are relevant and worth being studied. From a theoretical perspective, however, the first type of errors is slightly more interesting. This first type of error can really only occur when two games are played concurrently. In contrast, observation errors can already be discussed within the classical framework of a single repeated game.

Changes: In our revised main text, we describe more clearly why we have implemented ‘imperfect recall’ the way we did. This discussion follows the arguments given above.

5. In the context of concurrent repeated games, your study assumes that outcomes in one game can influence decision-making in another, a phenomenon that could be likened to ‘crosstalk.’ How does the mechanism of decision linkage you've identified differ from the ‘crosstalk’ effect described by Reiter, Johannes G., et al. in ‘Crosstalk in concurrent repeated games impedes direct reciprocity and requires stronger levels of forgiveness,’ published in *Nature Communications* 9.1 (2018): 555? Could the authors provide a comparative analysis of these two concepts and discuss how they might interact or diverge within the framework of concurrent games?

Reply: That is a very good observation. Indeed, our different-partner treatment was directly inspired by the model of Reiter and colleagues. The concepts of crosstalk and game linkage are closely related. However, in their paper, crosstalk is an exogenous parameter. In their model, players mistakenly copy the action in one game onto the other game with a fixed probability. In our model we allow for strategies to freely evolve their desired level of “crosstalk” or linkage. In the same-partner treatment, such linkage can be strategic; players may have an incentive to link their behavior across games. In contrast, in the different-partners treatment, such forms of crosstalk are more readily interpreted as (probably unintended) spillovers from one game to another.

Changes: Already in the SI of our original submission, we compared our model to the paper by Reiter et al in quite some detail. At the suggestion of several reviewers, we now discuss these differences in the main text (together with the paper by Donahue et al). In short, the two papers by Reiter et al and Donahue et al have inspired some of our work. However, the present manuscript is the first attempt to discuss concurrent games systematically, within a single framework.

Reviewer #3

How to understand the evolution and emergence of cooperation has attracted widespread attention from a broad range of disciplines in recent years. Direct reciprocity is one of the core mechanisms enabling cooperation among unrelated individuals. Yet most of the works regarding direct reciprocity assume that individuals either only engage in one repeated game at a time or that they treat each game as independent. Indeed, in most real examples, individuals engage in several games concurrently. In concurrent games, the outcome of one interaction may affect how individuals subsequently behave in a different game. In this manuscript, the authors propose a theoretical framework of concurrent games. They assume that individuals repeatedly engage in two independent stage games, either with the same or with different partners and agents adapt the strategies over time according to an evolutionary learning process. The authors find that individuals often link their behaviors across the two games and the linkage is particularly pronounced when the different games take place among the same partners. They also study four constraints and heuristics that may conceivably affect behavior in concurrent games. The authors show that the linkage in the concurrent games can have both a positive or a negative effect on cooperation. Additionally, they conduct a behavioral experiment to further explore the theoretical results. However, their experimental results show that the overall effect of two concurrent games on cooperation tends to be negative.

I have enjoyed reading this paper. I have found it well-written and well-organized and the results are interesting. It is my opinion that this manuscript systematically investigates the dynamics of cooperation in concurrent games and makes a contribution to the field.

Reply: Thank you for this positive summary. We also appreciate the helpful feedback below.

However, I also have some following comments or questions on this work.

In this work, the authors only consider that individuals repeatedly engage in two independent stage games in the framework of concurrent games they proposed. Indeed, individuals engage in two or more games simultaneously in concurrent games. From this point, the authors can propose a more general framework for concurrent games. In addition, they can also show whether their results presented in this paper are still valid when individuals engage in more than two games simultaneously. Besides, in concurrent games individuals may engage in different types of social dilemma games, not only play the donation game, as shown in Fig. S4. The authors may have some more discussion about this point in the manuscript.

Reply: Thank you for this comment. We agree, in much our manuscript we focus on concurrent games in which individuals engage in two games at once. Alternatively, one could also consider an even more general framework in which individuals engage in, say, n games in parallel.

While this is an interesting extension, we have not explored such a setup for three reasons. The first reason is didactical: We felt that many readers will find a framework with two games easier to interpret. This framing also allows us to use intuitive names for the two games (the *high-benefit game* and the *low-benefit game*). The second reason is computational. Especially the different-partners treatment requires a considerable computational effort. To see why, note that in our setup, there are four players, each one engaged in two games with two actions each. It follows that the number of states (possible

outcomes of a single round) is equal to $2^{2 \times 4} = 256$. Therefore, to compute the players' payoffs, we need to compute eigenvectors of a 256×256 transition matrix (as defined by our SI Eq. [12]). Suppose now we keep our setup with four players, but now assume that players engage in three parallel games instead. In that case, we need to deal with $2^{3 \times 4} = 4,096$ states, and a $4,096 \times 4,096$ transition matrix. Such computations are still feasible, but we felt that the added insight might not be worth the effort. Third, we wanted the model to match the setup of our experiment. For the experiment, we opted to have the simplest non-trivial design, based on two parallel games only. For simplicity, the model considers the same setup.

Having said that, we believe the 2-game model already captures most of the interesting effects in concurrent games. Already in the setup with two games, linkage can play an important role. Moreover, already with two games, it is instructive to analyze the effect of certain cognitive biases and spillovers. A setup with three games might yield different quantitative results. But we felt that already a setup with two games can get at the most important conceptual insights.

Changes: While we believe an extension to the case of n games is worthwhile, for this paper we prefer to keep our framework simple and to only consider two games. However, in our revised Discussion, we mention the case of n games being played in parallel as an interesting direction for future work. Regarding the discussion of different types of social dilemmas, we completely agree with the reviewer. In fact, we have already done such simulations for our previous submission (see **Fig. S4**), but the respective results were somewhat hidden in the SI. Now, we describe them in more detail in the main text.

The authors mainly present the cooperation rates in high and low games, respectively. It is fine. However, when individuals engage in concurrent games, there exist the resulting cross-over and spill-over effects. Readers may be interested in such effects on the overall cooperation rate in concurrent games. From this point, the authors can present some corresponding results in the manuscript.

Reply: Indeed, in this study we report cooperation rates separately, for the high-benefit and for the low-benefit game. We understand it is tempting to compute an additional metric, the game's overall cooperation rate, by adding up the two individual cooperation rates. However, we were worried that such a presentation might be misleading. After all, cooperation in the high-benefit game is much more valuable (in terms of the players' welfare), compared to cooperation in the low-benefit game. For these reasons, we prefer a presentation in which results for each game are shown separately. We hope that even with this presentation, we can show the potential effects of cross-over effects. One example is **Figure 2e**. This figure shows that in the same-partner treatment, players learn to condition their behavior also on what happened in the *other* game.

The authors explore theoretically and experimentally how different strategic motives and spillovers jointly shape individual actions on concurrent games. Their theoretical results show that the linkage in concurrent games can have both a positive or a negative effect on cooperation, compared to the standard case of independent games. However, their behavioral experiment suggests the overall effect of two concurrent games on cooperation tends to be negative. I think that it is important to explain why such discrepancy of theoretical predictions and experimental results can happen.

Reply: This is a fair comment. In our results we highlight that the lower cooperation rates in both treatments are already present in the first round, indicating that reciprocal mechanisms cannot explain this effect. Whatever cognitive behavioral mechanisms is responsible for this effect, our theoretical model does not account for it. Further research might want to look into concurrent interactions over

more than two games to see if this effect scales with each extra interaction or remains constant at a lower level.

Changes: We contrast the theoretical and experimental results in more depth in the Discussion and also mention possible future research.

I note that the idea of concurrent games has been proposed in previous works (see Ref. 25 and Ref. 26). Meanwhile, Ref. 26 shows that the crosstalk effect in concurrent games impedes the maintenance of cooperation. From this point, the authors should point out the focus of this present study and importantly they should stress the differences between their work and Refs. 25 and 26. In this present version of the manuscript, however, the authors did not make a deeper comparison. The novelty and contribution of this work need to be justified.

Reply: We agree, both studies served as important motivations for our current manuscript. In short, the paper by Donahue et al (previously Ref. 25) looked at a setup similar to our same-partner treatment. However, that paper did not look at the effect of cognitive biases, nor did it contain any empirical data. The paper by Reiter et al (previously Ref. 26) looked at a setup in which a player's experiences with one opponent can shape her behavior towards a different opponent. While the basic idea is similar to our different-partners treatment, the modeling approach is quite different. Reiter et al assume there is some exogenous probability with which events in one interaction affect a player's behavior in a different interaction. In our setup, we allow the endogenous evolution of linkage.

More generally, we consider our framework to be a more systematic model of behavior in concurrent games. With our model, we can distinguish between strategic motives to link one's behavior across games (as in the same-partner treatment) from non-strategic motives (as in the different-partner treatment).

Changes: Already in the SI of our first submission, we compared our paper to Refs. 25 and 26 in quite some detail. However, we agree with the reviewer that this discussion is important enough to appear in the main text. For our revised manuscript, we thus expand on the novelty of our study and how it relates to both Refs. 25 and 26 in our Introduction.

Reviewer #4

This work titled "Dynamics of cooperation in concurrent games," extends the traditional evolutionary game theory by developing an innovative theoretical framework to examine cooperation dynamics in environments where individuals participate in multiple concurrent games. Through a synthesis of theoretical modeling and behavioral experiments, the research delves into the intricacies of direct reciprocity, transcending the conventional presupposition of treating games in isolation. Employing introspection dynamics, the authors simulate the strategic evolution of cooperation within a controlled experimental design that accounts for cognitive limitations and heuristic-driven learning processes. The investigation uncovers that cross-game linkages can exert a bidirectional influence on cooperative behaviors, potentially enhancing or undermining the propensity for collaboration. The findings reveal a nuanced relationship between concurrent game engagement and cooperative outcomes, with significant implications for economic, psychological, and sociological perspectives on human interaction. This work advances our grasp of the factors that catalyze or impede the emergence of cooperative strategies in complex social settings.

Reply: Thank you for this summary and the positive feedback.

I have taken note of the authors' investigation into the dynamics of cooperation in concurrent games, especially their findings regarding how individual behaviors can influence each other across different games and form cross-game linkages. These findings resonate with the conclusions of the authors' previous work "Evolution of cooperation in stochastic games" (Nature 559, 7713 (2018): 246-249). However, to ensure the novelty and substantial contribution of this paper to the existing literature, I recommend that the authors: Clearly delineate the differences and connections between this work and previous studies in terms of theoretical framework, experimental design, or data analysis.

Reply: This is a good point. In the paper mentioned by the reviewer, the authors explore what happens if the players' environment can change in time. For example, if both players cooperate, they might be more likely to interact in a high-benefit game in the next round. Conversely, if both of them defect, the environment might deteriorate, and players might face a game with a lower benefit of cooperation. Thus, similar to our setup, players can engage in a set of distinct games (e.g., a high-benefit game and a low-benefit game).

However, the overall research question addressed in the stochastic-games paper is fundamentally different. That paper explores how individuals act when they know their actions can impact their ecological or social environment. But at any given round, players only engage in a single game. Instead, our analysis of concurrent games asks how individuals act when they simultaneously engage in several games at once. This focus leads to a different set of questions: When would individuals learn to treat each of their ongoing games as independent? When would they have an incentive to strategically link their behavior across games? And how do these incentives change, depending on whether the different games are played with the same partner, or with different partners?

In addition, we note that our current paper contains several elements that are completely absent in the stochastic games paper. For example, our paper introduces a general framework to systematically explore the role of different cognitive limitations. Moreover, we complement our theoretical model with a behavioral experiment, explicitly designed to explore behavior in concurrent games.

Changes: We now briefly mention stochastic games in the main text. In addition, we provide a more detailed description in the SI, in our section on the relevant previous literature.

Elaborate on the innovative aspects of this study compared to previous work, especially the new cognitive constraints and learning heuristics factors introduced when considering concurrent games.

Reply: Thank you for this encouragement. Indeed, we believe our analysis of cognitive constraints and learning heuristics is one of the key innovations of this study. For example, one such cognitive constraint is what we have termed 'imperfect recall'. Here, we refer to situations in which players might confuse the outcome of one interaction with what happened in another. We believe that in practice, such instances of imperfect recall might be at least as important as 'trembling hands' (a form of noise that has got much more attention in the repeated games literature). However, in our opinion, it takes a framework of concurrent games to meaningfully analyze the spillovers introduced by imperfect recall. In this manuscript, we introduce such a framework.

Changes: We have expanded our discussion of cognitive constraints, in order to highlight the innovative aspects of our study. Thank you for this feedback!

Discuss how this study further validates or extends previous theoretical models through experimental methods, and what new insights these experimental results provide for understanding cooperative behavior in concurrent games.

Reply and changes: In line with your comments and suggestions and that of the other reviewers, we have expanded our introduction and discussion. Specifically, we explain in more details how our study differs from previous ones and its novelty, discuss how the experimental results complement and support our theoretical results. In addition, our SI contains a more thorough literature review in which we discuss the relevant earlier work in full detail.